# Tocotrienol as a Protecting Agent against Glucocorticoid-Induced Osteoporosis: A Mini Review of Potential Mechanisms

**DOI:** 10.3390/molecules27185862

**Published:** 2022-09-09

**Authors:** Sophia Ogechi Ekeuku, Elvy Suhana Mohd Ramli, Norfarahin Abdullah Sani, Norzana Abd Ghafar, Ima Nirwana Soelaiman, Kok-Yong Chin

**Affiliations:** 1Department of Pharmacology, Universiti Kebangsaan Malaysia Medical Centre, Kuala Lumpur 56000, Malaysia; 2Department of Anatomy, Universiti Kebangsaan Malaysia Medical Centre, Kuala Lumpur 56000, Malaysia

**Keywords:** antioxidant, anti-inflammatory, bone, osteoblasts, osteoporosis, steroids

## Abstract

Glucocorticoid-induced osteogenic dysfunction is the main pathologyical mechanism underlying the development of glucocorticoid-induced osteoporosis. Glucocorticoids promote adipogenic differentiation and osteoblast apoptosis through various pathways. Various ongoing studies are exploring the potential of natural products in preventing glucocorticoid-induced osteoporosis. Preclinical studies have consistently shown the bone protective effects of tocotrienol through its antioxidant and anabolic effects. This review aims to summarise the potential mechanisms of tocotrienol in preventing glucocorticoid-induced osteoporosis based on existing in vivo and in vitro evidence. The current literature showed that tocotrienol prevents oxidative damage on osteoblasts exposed to high levels of glucocorticoids. Tocotrienol reduces lipid peroxidation and increases oxidative stress enzyme activities. The reduction in oxidative stress protects the osteoblasts and preserves the bone microstructure and biomechanical strength of glucocorticoid-treated animals. In other animal models, tocotrienol has been shown to activate the Wnt/β-catenin pathway and lower the RANKL/OPG ratio, which are the targets of glucocorticoids. In conclusion, tocotrienol enhances osteogenic differentiation and bone formation in glucocorticoid-treated osteoblasts while improving structural integrity in glucocorticoid-treated rats. This is achieved by preventing oxidative stress and osteoblast apoptosis. However, these preclinical results should be validated in a randomised controlled trial.

## 1. Introduction

Osteoblasts are specialised bone-forming cells that play many roles in bone remodelling. Osteoblasts develop from multipotent mesenchymal cells under the influence of regulatory transcription factors runt-related transcription factor 2 (RUNX2) and osterix, which also can grow into other cell lineages such as adipocytes, myocytes and chondrocytes [1]. Osteoblasts produce organic bone matrix (osteoid) and subsequently mineralise it during skeletal modelling and bone remodelling [2]. The differentiation and function of osteoblasts are regulated by several signalling pathways, one of which is the canonical Wingless and Int-1 (Wnt) signalling pathway. It is activated by the binding of Wnt1 and Wnt3a protein with Fizzled and low-density lipoprotein receptor-related protein 5/6. This leads to the inhibition of glycogen synthase kinase-3 beta, which phosphorylates cytoplasmic beta-catenin, causing its degradation. The accumulation of beta-catenin leads to its nuclear translocation, forming a complex with T-cell factor/lymphocyte enhancer factor 1 and cAMP response element-binding-binding protein and translation of genes responsible for osteoblast differentiation and bone formation. Sclerostin (SOST) and dickkopf-1 (DKK1) are inhibitors of the Wnt signalling pathway secreted by osteocytes [3,4]. Osteoblasts also secrete receptor activator of nuclear factor kappa-Β (RANK) ligand (RANKL), which binds with RANK on osteoclasts precursors and stimulates their differentiation. At the same time, osteoblasts also secrete osteoprotegerin (OPG), which is a decoy receptor for RANKL, to prevent the binding of RANKL with RANK and suppress osteoclast formation. These signalling pathways are influenced by various endogenous and exogenous factors, including inflammatory cytokines, oxidative stress and glucocorticoids (GCs) [5].

GCs are widely used as the treatment for inflammatory diseases and as chemotherapeutic agents. They are crucial for the induction of osteoblast differentiation and the formation of a mineralised extracellular matrix, as they influence the signals coordinating the differentiation of multipotent mesenchymal precursors cells into osteoblasts [6,7,8]. GCs have been reported to stimulate the expression of differentiation markers of both osteoblasts and adipocytes in in vitro studies [9]. However, the prolonged use of GCs leads to osteoporosis. High GC levels have been reported to suppress osteoblastic differentiation and function [10] and induce osteoblast apoptosis, leading to the suppression of bone formation [11]. The effects of high GC on osteoblast are mainly mediated by suppression of the Wnt signalling pathway [12].

The current prevention for GC-induced osteoporosis (GIO) is ensuring an adequate intake of vitamin D and calcium among patients on GCs. The assessment of fall risk in patients taking high doses of GCs should be performed. Lifestyle modifications similar to those adopted by patients with postmenopausal osteoporosis, which include weight-bearing exercise and the elimination of other risk factors, such as smoking or alcohol, should also be implemented by chronic GC users [13]. Patients using GCs at moderate or high osteoporosis risk are recommended to start with pharmacologic agents. The first-line agent is bisphosphonate therapy due to its efficacy, safety and low cost. Bisphosphonates, such as once-weekly oral alendronate or once-monthly oral ibandronate, have been the most studied agents for the prevention and treatment of GIO. Treatment with bisphosphonates has been proven to improve bone mineral density (BMD) or vertebral fracture risk and reduced the hip fracture risk [14]. If bisphosphonates are contraindicated, teriparatide is suggested as the second-line agent, which is followed by denosumab. Teriparatide, an anabolic agent, has also been shown to reduce fracture risk [15]. Denosumab, a receptor activator of nuclear factor kappa-Β (RANK) ligand (RANKL) inhibitor, has been reported to increase lumbar spine and hip BMD. However, the data regarding its efficacy and safety are still very limited [16]. The search for other alternative agents to preserve skeletal health in GC users is still ongoing.

Vitamin E encompasses both tocotrienol and tocopherol, both of which are potent antioxidants. They could be divided into alpha (α), beta (β), gamma (γ) and delta (δ) isomers. Tocotrienol and tocopherol share a similar structure, which consists of a chromanol ring and a long carbon tail. Tocotrienols have three double bonds, whereas tocopherols have only single bonds on the carbon tail (Figure 1) [17]. The main sources of tocotrienol include rice bran, palm, and annatto. Palm-derived tocotrienol (PTT) contains approximately 75% tocotrienol and 25% tocopherol, whereas annatto-derived tocotrienol (ATT) contains solely tocotrienol (99.9%) [18]. Previous studies have established the role of tocotrienol as an anti-oxidative, anti-inflammatory, anti-hyperlipidaemic and anti-hyperglycaemic agent [18,19,20,21]. Many studies also suggested that tocotrienol can prevent or reverse bone loss [18,19,20]. Thus, tocotrienol could potentially reverse bone loss induced by GCs.

This review aims to summarise the preclinical evidence of the skeletal effects of tocotrienol in GIO models. The potential mechanisms of tocotrienol in achieving bone protection under the influence of GC are also discussed.

## 2. Glucocorticoids and Bone

Before appraising the effects of tocotrienol in preventing GIO, it is necessary to understand the actions of GCs on bone cells. The information will allow us to understand the avenues of intervention in GIO.

### 2.1. Direct Action of Glucocorticoid on Osteoblasts

GC-induced osteoblast dysfunction is one of the main pathological mechanisms underlying the development of GIO [23]. GCs impair osteoblast proliferation, increase apoptosis and alter autophagy through changing RANKL/OPG and Wnt/sclerostin expression. Their inhibitors, microRNAs, IL-11, BMP/notch signalling, and effectors of apoptosis also play a major role in the action of GCs on the osteoblasts [24]. GCs suppress the canonical Wnt signal pathway that stimulates osteoblast differentiation and activity, partially through the enhancement of the DKK1 production in cultured human osteoblasts [25]. Using dexamethasone (DEX) as an example, at higher concentrations, it inhibits osteoblast differentiation by decreasing alkaline phosphatase (ALP) activity, RUNX2 and osteocalcin (OCN) expressions, and increasing RANK expression [25,26].

The actions of GCs on bone are also determined by variations in the expression and sensitivity of the GC receptors, the export of steroids from the cell by transmembrane transporters, and enzymatic metabolism of GCs to the more or less active metabolites [27]. 11β-hydroxysteroid dehydrogenases (11β-HSDs), which control the interconversion between the active cortisol and corticosterone and their inactive counterparts, cortisone and dehydrocorticosterone, also contribute to the skeletal action of GC on bone [27]. All these events contribute to increased bone turnover, reduced mineralization, and subsequently, bone loss.

GCs are important in the differentiation of osteoblasts, but their effects on osteoblast proliferation are inconsistent. At lower doses (10^−7^M DEX), GCs promote uncommitted mesenchymal precursors cells to differentiate into osteoblasts [28,29]. However, at high doses (>10^−7^M DEX), GCs inhibit the proliferation of osteoblast-like cells in culture [30]. GCs decrease bone formation in most in vivo studies but stimulate it in in vitro studies. These opposing effects could be due to low levels of GCs being stimulatory at lower concentrations and inhibitory at higher concentrations [31].

Multipotent mesenchymal precursor cells could develop into osteoblasts, adipocytes or other cell lineages. GCs facilitate the commitment of these precursors cells to differentiate into certain cell types [32]. GCs stimulate the expression of markers of differentiation of both osteoblasts and adipocytes in in vitro studies [33]. In animal and human studies, GCs are associated with bone marrow adiposity within the femoral neck [34,35]. Genetic deletion of 11β-HSD1 and the lowering of GC levels were reported to affect marrow adipose tissue but not bone formation [36].

GCs upregulate DKK1 expressed by the osteoblasts, which in turn suppress anabolic osteoblast behavior [37]. GCs also stimulate osteoblast apoptosis in vitro via the increased endoplasmic reticulum stress through a synergistic pathway with TNFα [38]. This triggers the rapid activation of the kinases Pyk2 and JNK and increases reactive oxygen species in primary cultures [39,40]. The Bcl2 family such as Bim is the proapoptotic factor, and its expression is regulated by GCs [41]. GCs also upregulate Bak expression and downregulate Bcl-xL expression (prosurvival) [42]. DEX can also induce Bcl2-mediated cell death via the induction of p53 [43].

Insulin growth factors (IGFs), transforming growth factors (TGFs), fibroblast growth factors (FGF), and platelet-derived growth factors are also other signalling pathways also targeted by GCs. GCs reduce the anabolic actions of TGFβ and suppress the expression of IGF-I and platelet-derived growth factors, which possess anabolic mitogenic actions in osteoblasts [44,45,46]. Novel cellular targets in osteoblasts that are influenced by GCs in vitro include interleukin (IL)-11 [10], E3 ubiquitin ligases [47], and microRNA-199a [30]. The suppression of IL-11 has been identified as an important mediator of the adverse effects GCs on bone [10]. Therapeutic GCs reduce sex steroid levels [48], which is likely to impact patients with serious inflammatory illnesses greater due to the effects of inflammation on the hypothalamic–pituitary–gonadal axis. This cascade will be harmful to the bone [49]. GC excess leads to central fat accumulation due to its impacts on fat metabolism, which is associated with increased insulin resistance. Obesity and diabetes both have complex impacts on bone metabolism and fracture risk [50,51].

### 2.2. Action of Glucocorticoid on Oxidative Stress in Osteoblasts

GCs can cause oxidative stress through the production of reactive oxygen species (ROS), downregulation of cytoprotective antioxidant proteins and antioxidant enzyme activities [52]. High ROS levels inhibit osteoblast differentiation and function, causing osteoblast cell death and growth reduction [53,54]. DEX treatment induces oxidative damage by depleting total antioxidant capacity while increasing ROS formation and lipid peroxidation. It also causes a significant reduction in the RUNX2 mRNA expression, which underlies high-dose DEX-induced osteotoxicity [55]. DEX treatment also triggers a significant decline in the mitochondrial membrane potential due to upregulated caspase activity. Treatment with antioxidants can upregulate the expression levels of these osteogenic markers and downregulate caspase expression, thus decreasing the apoptotic effect of DEX. This observation suggests the involvement of oxidative stress in DEX-induced osteoporosis [54].

A study by Zhang et al. [17] found that DEX-treated MC3T3-E1 pre-osteoblast cells express lower nuclear factor (erythroid-derived 2)-like 2 (Nrf-2) and their target proteins with a decline in oxidative stress markers. This negative impact could be reversed with plumbagin, which is an antioxidant. Nrf-2 is a transcriptional activator, which binds to antioxidant responsive element (ARE) and enhances the expression of antioxidant enzymes. This endogenous antioxidant defence is activated in a cellular oxidative stress event. ROS also induces endoplasmic reticulum stress and autophagy-mediated apoptosis [56]. A study by Liu et al. demonstrated that DEX induces apoptosis, endoplasmic reticulum stress, ROS formation and autophagy in pre-osteoblasts [57]. Animal studies demonstrated that high-dose DEX treatment reduced bone generation and destroyed bone trabeculae in rats, leading to microarchitectural degenerative changes mimicking osteoporosis [58,59]. Therefore, inhibiting the production of ROS may provide an avenue of intervention against GC-induced apoptosis of MC3T3-E1 cells [57].

Most in vivo studies showed that GCs usually cause a decrease in bone formation, but some in vitro studies showed largely stimulatory effects of GCs actions. This observation may suggest that a low GC level possesses stimulatory effects, while a high GC level possesses inhibitory effects on bone [10]. GCs exert anti-inflammatory effects on osteoblasts by suppressing cytokines, such as IL-11, via the interaction of the monomeric GR with AP-1 but not nuclear factor kappa B (NF-κB). The inhibition of cytokines by GCs attenuates osteoblast differentiation, which partly accounts for bone loss during GC therapy [60].

### 2.3. Action of Glucocorticoid on Osteoclasts

GCs create an environment favouring osteoclast formation and bone resorption activities by increasing RANKL and suppressing OPG secretion by osteoblasts [61,62,63,64]. This process may be mediated by miR-17/20a in osteoblasts and miR-182 in osteoclasts [65,66]. GCs also affect osteoclast functions directly. GCs can induce osteoclast-mediated bone resorption without affecting their apoptosis rate, and this process requires the dimeric GC receptor [67]. GCs can improve autophagy in osteoclasts and promote their survival through the PI3K/Akt/mTOR signalling pathway [68]. GCs can affect the geometry of osteoclast resorption activities by forming more trench-like resorption pits, which directly affect bone stiffness, with the lumbar as the most affected bone site [69]. At a similar exposure level, GCs can induce mitochondria dysfunction and oxidative stress in osteoblasts but not osteoclasts [70]; this may contribute to the imbalanced bone remodelling observed in GIO. However, prolonged GC exposure may be destructive to osteoclasts and their functions, hindering bone remodelling cycle and predisposing users to osteoporosis [71].

## 3. Tocotrienol and Bone

Tocotrienol has been reported to demonstrate potential skeletal anabolic effects in in vivo and in vitro studies. In an earlier study, it was found that PTT preserved the bone calcium content in DEX-treated adrenalectomized rats. More recent studies on oestrogen and androgen-deficient rat models reported tocotrienol supplementation preserved the bone calcium content, and this was parallel to the observed cortical bone thickening at the midshaft caused, which also corresponded to the increased bone mechanical strength [72,73]. Another study on osteoporosis induced by buserelin, a gonadotropin-releasing hormone agonist, also demonstrated similar effects of ATT. The skeletal protective effects of ATT were comparable to calcium supplementation [73].

The efficacy of two different natural mixtures of tocotrienol, i.e., ATT and PTT, has been compared in adrenalectomised rats supplemented with DEX. Both tocotrienol mixtures were reported to maintain osteoblast surface in the femoral trabecular bone of GC-treated rats [58,59]. The decrease in the osteoblast number and osteoblast dysfunction leads to a reduction in the synthesis of bone matrix [74]. Thus, ATT and PTT could preserve osteoblast survival and function, thereby preserving skeletal integrity.

The differentiation and proliferation of osteoblasts are regulated by cascades of genes [8,75]. GC was reported to upregulate the expression of genes coding for type I collagen and osteocalcin [74,76]. This could be a compensatory action to fill the resorbed cavities because bone resorption and formation are coupled [77]. ATT supplementation significantly decreased the mRNA expression of collagen type 1 and osteocalcin in the femur of rats with GC-induced osteoporosis [59]. Supplementation with PTT also reduced the expression of genes coding for osteocalcin and collagen 1 alpha 1. These results suggest that ATT and PTT could suppress the high bone remodelling phenotype commonly observed with osteoporosis. ATT also increased the osterix expression in the femur in GC-induced osteoporotic rats [58], suggesting that ATT might improve osteoblast differentiation and maturation [78].

Correcting oxidative stress induced by GCs using an antioxidant agent might be a viable method to prevent osteoporosis. The impact of vitamins and antioxidants on bone healing has been reported to have a positive effect on preventing osteoporosis. As oxidative stress is proven to be toxic to osteoblasts, agents that reduce oxidative stress may be beneficial in preserving bone formation. Tocotrienol has been shown to prevent oxidative stress-related diseases effectively [18,19,20,21,79]. PTT and ATT supplementation (60 mg/kg/day) significantly reduced lipid peroxidation marked by malondialdehyde levels as well as increasing antioxidant defence marked by superoxide dismutase and glutathione peroxidase activity in the bones of DEX-treated rats [58,59]. These effects might reduce the toxic effects of oxidative stress on the osteoblasts, which translates to preserved osteoblasts surface in the ATT-supplemented rats compared to the non-supplemented rats [59].

Apart from that, tocotrienol might enhance bone formation directly. PTT and ATT increased bone dynamic parameters (bone mineralized surface, mineral apposition rate and bone formation rate) in rats with GIO [58,59]. The preservation of osteoblasts and their bone formation activities could maintain the structural integrity of the bone. As evidence, PTT and ATT treatment increased bone volume, trabecular number and thickness while decreasing trabecular separation in rats with GC-induced osteoporosis [58,59]. With the preservation of bone structural integrity, bone biomechanical strength was also preserved.

The RANKL/OPG and Wnt/β-catenin pathway are important targets of GCs. Apart from the positive effects of ATT and PTT in combating the effects of GCs on bone, they were also found to have protective effects in the oestrogen and androgen-deficient rat models as well as in osteoporosis induced by metabolic syndrome [5]. ATT (unformulated or self-emulsified) was found to improve bone mineralisation and osteoblast number in rats with prolonged oestrogen deficiency by reducing the sclerostin level and RANKL/OPG ratio in the bone, which caused a rise in the bone formation and osteoblastic activity [80]. Suppression of the sclerostin level in the ovariectomised was suggestive of the involvement Wnt/β-catenin pathway in the skeletal action of ATT. Another study conducted by Chin et al. observed an increase in the mRNA expression of beta-catenin in bone tissue from the androgen-deficient rat model. This study also reported an increased expression of genes coding for ALP, collagen type I alpha 1 and osteopontin and a reduction in RANKL in the orchidectomised rats [81].

Another study on osteoporosis caused by metabolic syndrome showed that ATT lowered RANKL, sclerostin, DKK-1 and FGF-23 levels, which prevented the deterioration of trabecular bone microstructure and increased the ultimate load, Young’s modulus of elasticity, and decreased the ultimate strain of femur in the high-carbohydrate high-fat (HCHF) rats [82,83]. Apart from that, the administration of ATT at 60 or 100 mg/kg reversed the elevated IL-1α and IL-6 levels in the HCHF animals and normalised the TNF-α [82]. The high leptin level that resulted from obesity stimulates an inflammatory response to induce bone loss through the activation of the RANK/RANKL/OPG pathway [84,85]. ATT at 100 mg/kg significantly reduced leptin levels in the HCHF animals, which contributed to the inhibition of bone loss [86]. ATT at the dose of 60 or 100 mg/kg also significantly increased adiponectin levels in the HCHF animals [86]. Adiponectin upregulated the expression of type II collagen, RUNX2, and increased ALP activities to increase chondrocyte proliferation, proteoglycan synthesis and matrix mineralisation [87]. Moreover, adiponectin enhanced BMP-2 expression in osteoblasts via the AdipoR receptor signalling pathway [88]. An in vitro study reported that 50 μM of α-tocotrienol suppressed RANKL expression in osteoblasts and RANKL induced the expression of c-Fos and NFATc1 by inhibiting extracellular-signal-regulated kinase (ERK) and NF-κB activation [89]. Both tocotrienols also prevented the increase in sclerostin and DKK-1 levels in the animals with metabolic syndrome, which negatively regulate the activation of Wnt/β-catenin by preventing the binding of Wnt ligands to the frizzled receptor, lipoprotein receptor-related protein 5/6 [90].

The direct effects of tocotrienol on GC-treated osteoclast formation and function have not been studied in depth. From other studies, tocotrienol isomers directly inhibited tartrate-resistant acid phosphatase-positive cell formation from peripheral blood mononuclear cells and bone marrow macrophages without affecting cell survival [89,91]. Tocotrienols also prevented RANKL expression in osteoblasts and contribute to less osteoclast formation in a coculture of osteoblasts and bone marrow cells [89]. The suppression effects of tocotrienols might be mediated by the MAPK pathway at the early stage and NF-κB pathway at the later stage [89]. However, these studies were not conducted in the presence of GCs; hence, the effects of tocotrienol on osteoclast differentiation, activity and survival under the influence of GCs are yet to be confirmed.

In summary, the research available so far suggests that tocotrienol protects bone from the effects of GCs by preventing oxidative stress and apoptosis and increasing the differentiation and maturation in osteoblasts (Figure 2).

The effects of tocotrienol in GIO have not been examined in humans. There are only two studies on the protective effects of tocotrienol on bone health in humans. The effects of ATT in protecting bone health have been tested among postmenopausal osteopenic women. They were randomised into groups receiving placebo/olive oil, low-dose (430 mg) and high-dose ATT (860 mg) for 12 weeks. There was a significant reduction in urine N-telopeptides (NTX) levels, soluble RANKL, soluble RANKL/OPG ratio and urine 8-hydroxy-2′-deoxyguanosine and an improvement in bone ALP/NTX ratio. However, serum OPG and urinary calcium levels were not affected by the supplementation. No dose-dependent effects were observed between the two doses. BMD values were not evaluated in this study due to the short duration of action [92]. In a related study, ATT supplementation (600 mg) for 12 weeks increased lysophospholipids but reduced acylcarnitines and catabolites of tryptophan and steroids in postmenopausal, osteopenic women, indicating a suppression of inflammation and oxidative stress that could be beneficial to bone health [93]. Throughout the 12-week study period, ATT up to 600 mg did not affect liver or kidney functions in postmenopausal women with osteopenia. The subjects also did not report adverse effects during the study period [94]. However, these studies only infer the general protective effects of tocotrienol on bone health but not particularly on GIO. It should be noted that the subjects did not have established osteoporosis, so it cannot be ascertained that tocotrienol can reverse bone loss. The efficacy of tocotrienol has not been compared against standard osteoporosis medications, such as vitamin D and calcium, bisphosphonate or teriparatide. All the available studies used ATT rich in delta-tocotrienol, but none had used palm tocotrienol with a full spectrum of tocotrienol isomers and α-tocopherol. Given that bone metabolism and weight-bearing mechanisms are different between humans and rodents, a properly planned human study on the effects of tocotrienol on GIO should be conducted.

## 4. Conclusions

Given the current evidence, there is a potential that tocotrienol will be able to reverse the negative effects of GCs on bone health by preserving osteoblasts. However, no randomized controlled trial has been conducted to validate the preclinical results currently. Efforts should be invested in bridging this gap so that tocotrienol could be used to reduce the skeletal outcomes among patients using GCs.

## Figures and Tables

**Figure 1 molecules-27-05862-f001:**
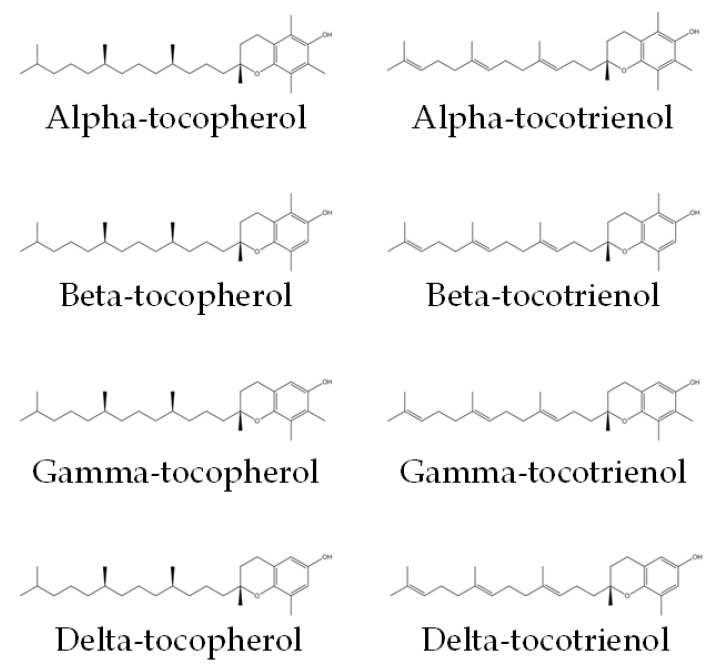
Chemical structures of tocopherols and tocotrienols [22]. They consist of a chromanol ring and a long carbon tail. On the carbon tail, tocotrienols have three double bonds, whereas tocopherols have only single bonds.

**Figure 2 molecules-27-05862-f002:**
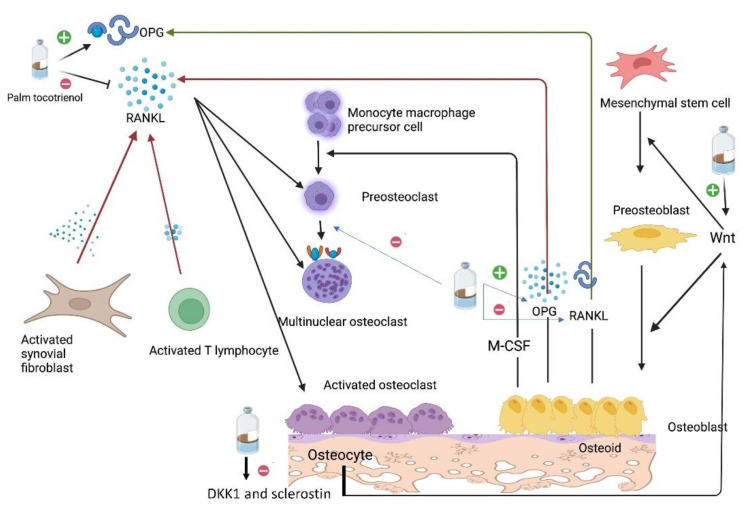
The postulated mechanism of tocotrienol in protecting against glucocorticoid-induced osteoporosis. Tocotrienol can increase OPG secretion by osteoblasts and reduce RANKL secretion by osteoblasts, fibroblasts and immune cells, thus reducing osteoclasts formation. Tocotrienol may also suppress osteoclast formation directly. In addition, tocotrienol can reduce DKK1 and sclerostin levels; thus, the Wnt signalling pathway and osteogenesis are not inhibited.

## Data Availability

Not applicable.

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
