# Peer review of "Tocotrienol as a Protecting Agent against Glucocorticoid-Induced Osteoporosis: A Mini Review of Potential Mechanisms"

_molecules, 2022, doi:10.3390/molecules27185862_

Round 1

Reviewer 1 Report

   Authors summarized the potential mechanisms of tocotrienol as a protecting agent to prevent glucocorticoid-induced osteoporosis. The review is extensive and in detail. Comments are as the followings.

1. Readers, especially the clinical practitioners, will definitely like to know if there is any evidence or experience of clinical use in human. Authors had provided many action mechanisms, almost solely in animal studies. Everyone in medical field knows that the effect/dose/AEs... will be very different between study animal and human. This article will be more persuasive if authors can search to provide human evidence of experience.

2. Glucocorticoid-related BMD reduction is certainly generalized, but it is more noticeable over femoral neck and spines. Most of the studies in the reference list are perhaps from rat studies. Human BMD reduction and osteoporosis can be very difference from those in rats, because of the great difference of weight-bearing pattern. Therefore, readers will wonder if the studied mechanisms can also apply to human. I wonder if authors can find any studies to address this issue, since the present article did not provide any human data.

Author Response

Comments:  

Authors summarized the potential mechanisms of tocotrienol as a protecting agent to prevent glucocorticoid-induced osteoporosis. The review is extensive and in detail. Comments are as the followings.

  1. Readers, especially the clinical practitioners, will definitely like to know if there is any evidence or experience of clinical use in human. Authors had provided many action mechanisms, almost solely in animal studies. Everyone in medical field knows that the effect/dose/AEs... will be very different between study animal and human. This article will be more persuasive if authors can search to provide human evidence of experience.
  2. Glucocorticoid-related BMD reduction is certainly generalized, but it is more noticeable over femoral neck and spines. Most of the studies in the reference list are perhaps from rat studies. Human BMD reduction and osteoporosis can be very difference from those in rats, because of the great difference of weight-bearing pattern. Therefore, readers will wonder if the studied mechanisms can also apply to human. I wonder if authors can find any studies to address this issue, since the present article did not provide any human data.

Reply:

Thank you for the comments. Since both points are connected, please allow us to respond to them together.

We acknowledge that data on the efficacy of tocotrienol among humans with osteoporosis/osteopenia are very limited in general. Considering that, we have added the existing literature to the manuscript as the following:  

Line 304-328: The effects of tocotrienol in GIO have not been examined in humans. There are only two studies on the protective effects of tocotrienol on bone health in humans. The effects of annatto tocotrienol in protecting bone health have been tested among post-menopausal osteopenic women. They were randomised into groups receiving place-bo/olive oil, low-dose (430 mg) and high-dose annatto tocotrienol (860 mg) for 12 weeks. Significant reduction in urine N-telopeptides (NTX) levels, soluble RANKL, soluble RANKL/OPG ratio and urine 8-hydroxy-2'-deoxyguanosine and an improvement in bone ALP/NTX ratio. However, serum OPG and urinary calcium levels were not affected by the supplementation. No dose-dependent effects were observed between the two doses. BMD values were not evaluated in this study due to the short duration of action [92]. In a related study, annatto tocotrienol supplementation (600 mg) for 12 weeks increased lysophospholipids but reduced acylcarnitines and catabolites of tryptophan and steroids in postmenopausal, osteopenic women, indicating suppression of inflammation and oxidative stress that could be beneficial to bone health [93]. Throughout the 12-week study period, annatto tocotrienol up to 600 mg did not affect liver or kidney functions in postmenopausal women with osteopenia. The subjects also did not report adverse effects during the study period [94]. However, these studies only infer the general protective effects of tocotrienol on bone health, but not particularly on GIO. It should be noted that the subjects did not have established osteoporosis, so it cannot be ascertained that tocotrienol can reverse bone loss. The efficacy of tocotrienol has not been compared against standard osteoporosis medications, such as vitamin D and calcium, bisphosphonate or teriparatide. All the available studies used annatto tocotrienol rich in delta-tocotrienol, but none had used palm tocotrienol with a full spectrum of tocotrienol isomers and α-tocopherol. Given that bone metabolism and weight-bearing mechanism are different between humans and rodents, a properly planned human study on the effects of tocotrienol on GIO should be conducted. 

Reviewer 2 Report

The manuscript summarizes interesting knowledge on the effects of tocotrienol on glucocorticoid-induced osteoporosis and discusses potential mechanisms of the effects. However, please reconsider my concerns listed below that should be further improved.

Major Concerns:

1. Regarding the description of the authors on glucocorticoid and bone, the authors explained the latest knowledge on the direct action of glucocorticoid action on osteoblasts. However, why did the authors avoid the explanation of the direct action of glucocorticoid action on osteoclasts and bone cells in the current manuscript? To improve the current manuscript, please more clearly and concisely describe this point in the revised manuscript.

2. Regarding the description of the authors on tocotrienol and bone, the authors explained the updated knowledge on the direct action of glucocorticoid action on osteoblasts. However, there is no explanation of the direct action of glucocorticoid action on osteoclasts and bone cells in the present manuscript. To improve the current manuscript, please add a concise description regarding this point in the revised manuscript.

3. In accordance with the above revisions, please entirely re-edit the present manuscript. 

Minor Concerns:

1. To increase the readability of the present manuscript, I recommend the authors insert the figure that explains the chemical structural formula of tocotrienol described in the manuscript.

2. Please more kindly explain each technical term in the main text for the readers outside of disciplines where the terms firstly appear or in the Introduction section. For example, what is dickkopf-1 (page 2, line 94)? In addition, what is the function of sclerostin (page5, line 227)?

3. Although the manuscript type is Mini Review, the authors should insert kind and concise explanations of the intracellular signaling pathways used in the present manuscript into the Introduction section for readers outside of disciplines.

4. Please more concretely explain the meaning of at high dose (page 3, line 108).

5. Is the word AnTT (page 4, line 185) just a misdescription? Please check.

6. Please improve the legend for Figure 1 by more kindly explaining the meaning of the figure. The current description is not enough at all, for the readers understanding of the meaning of the figure.

7. In accordance with the above major revision, I recommend the authors update Figure 1 itself if needed.

Author Response

Comment:

The manuscript summarizes interesting knowledge on the effects of tocotrienol on glucocorticoid-induced osteoporosis and discusses potential mechanisms of the effects. However, please reconsider my concerns listed below that should be further improved.

Major Concerns:

Regarding the description of the authors on glucocorticoid and bone, the authors explained the latest knowledge on the direct action of glucocorticoid action on osteoblasts. However, why did the authors avoid the explanation of the direct action of glucocorticoid action on osteoclasts and bone cells in the current manuscript? To improve the current manuscript, please more clearly and concisely describe this point in the revised manuscript.

Reply: Thank you for the comment. We have added the actions of glucocorticoids on osteoclasts in the manuscript as the following:

Line 193-205: GCs create an environment favouring osteoclast formation and bone resorption activities by increasing RANKL and suppressing OPG secretion by osteoblasts [61-64]. This process may be mediated by miR-17/20a in osteoblasts and miR-182 in osteoclasts [65, 66]. Besides, GCs also affect osteoclast functions directly. GCs can induce osteoclast-mediated bone resorption without affecting their apoptosis rate, and this process requires the dimeric GC receptor [67]. GCs can improve autophagy in osteoclasts and promote their survival through PI3K/Akt/mTOR signalling pathway [68]. GCs can affect the geometry of osteoclast resorption activities by forming more trench-like re-sorption pits, which directly affect bone stiffness, with the lumbar as the most affected bone site [69]. At a similar exposure level, GCs can induce mitochondria dysfunction and oxidative stress in osteoblasts but not osteoclasts [70], this may contribute to the imbalanced bone remodelling observed in GIO. However, prolonged GC exposure may be destructive to osteoclasts and their functions, hindering the bone remodelling cycle and predisposes users to osteoporosis [71].

Comment: Regarding the description of the authors on tocotrienol and bone, the authors explained the updated knowledge on the direct action of glucocorticoid action on osteoblasts. However, there is no explanation of the direct action of glucocorticoid action on osteoclasts and bone cells in the present manuscript. To improve the current manuscript, please add a concise description regarding this point in the revised manuscript.

Reply: Thank you for the comments. There is no study on the effects of tocotrienol on osteoclasts stimulated with glucocorticoids. However, we have added the available studies of tocotrienol on osteoclasts in the manuscript as the following:

Line 285-294: The direct effects of tocotrienol on GC-treated osteoclast formation and function have not been studied in depth. From other studies, tocotrienol isomers directly inhibited tartrate-resistant acid phosphatase-positive cell formation from peripheral blood mononuclear cells and bone marrow macrophages without affecting cell survival [89, 91]. Tocotrienols also prevented RANKL expression in osteoblasts and contribute to less osteoclast formation in a coculture of osteoblasts and bone marrow cells [89]. The suppression effects of tocotrienols might be mediated by the MAPK pathway at the early stage and NF-κB pathway at the later stage [89]. However, these studies were not conducted in the presence of GCs, hence the effects of tocotrienol on osteoclast differentiation, activity and survival under the influence of GCs are yet to be confirmed.

Comment: To increase the readability of the present manuscript, I recommend the authors insert the figure that explains the chemical structural formula of tocotrienol described in the manuscript.

Reply: Thank you for the suggestion. We have added the chemical structures of tocotrienols and tocopherols as Figure 1 in the manuscript.

Comment: 

Please more kindly explain each technical term in the main text for the readers outside of disciplines where the terms firstly appear or in the Introduction section. For example, what is dickkopf-1 (page 2, line 94)? In addition, what is the function of sclerostin (page5, line 227)?

Although the manuscript type is Mini Review, the authors should insert kind and concise explanations of the intracellular signaling pathways used in the present manuscript into the Introduction section for readers outside of disciplines.

Reply:

Thank you for the suggestions. Since these two points are connected, allow us to address them together. We have added the description of the two most relevant pathways in bone remodelling in the manuscript and explained the function of sclerostin and dickkopf-1.

Line 37-53: The differentiation and function of osteoblasts are regulated by several signalling pathways, one of which is the canonical Wingless and Int-1 (Wnt) signalling pathway. It is activated by the binding of Wnt1 and Wnt3a protein with Fizzled and low-density lipoprotein receptor-related protein 5/6. This leads to the inhibition of glycogen synthase kinase-3 beta, which phosphorylates cytoplasmic beta-catenin, causing its degradation. The accumulation of beta-catenin leads to its nuclear translocation, forming a complex with T-cell factor/lymphocyte enhancer factor 1 and cAMP response element-binding-binding protein and translation of genes responsible for osteoblast differentiation and bone formation. Sclerostin (SOST) and dickkopf-1 (DKK1) are inhibitors of the Wnt signalling pathway secreted by osteocytes [3, 4]. Osteoblasts also secrete receptor activator of nuclear factor kappa-Β (RANK) ligand (RANKL), which binds with RANK on osteoclasts precursors and stimulates their differentiation. At the same time, osteoblasts also secrete osteoprotegerin (OPG), which is a decoy receptor for RANKL, to prevent the binding of RANKL with RANK and suppress osteoclast formation. These signalling pathways are influenced by various endogenous and exogenous factors, including inflammatory cytokines, oxidative stress and glucocorticoids (GCs) [5].

Comment: Please more concretely explain the meaning of “at high dose” (page 3, line 108).

Reply: Thank you for the comment. We have added the exposure range that reflects the phrase “high doses” as the following:

Line 127: However, at high doses (>10-7M DEX), GCs inhibit the proliferation of osteoblast-like cells in culture.

Comment: Is the word “AnTT” (page 4, line 185) just a misdescription? Please check.

Reply: Thank you for the reminder. We have replaced “AnTT” with “ATT”. (Line 215)

Comment: Please improve the legend for Figure 1 by more kindly explaining the meaning of the figure. The current description is not enough at all, for the reader’s understanding of the meaning of the figure.

Reply: Thank you for the comment. We have edited the legend of Figure 2 (previously Figure 1) as the following:

Line 299-303: Figure 2. The postulated mechanism of tocotrienol in protecting against glucocorticoid-induced osteoporosis. Tocotrienol can increase OPG secretion by osteoblasts and reduce RANKL secretion by osteoblasts, fibroblasts and immune cells, thus reducing osteoclasts formation. Tocotrienol may also suppress osteoclast formation directly. Besides, tocotrienol can reduce DKK1 and sclerostin levels, thus Wnt signalling pathway and osteogenesis are not inhibited.

Comment: In accordance with the above major revision, I recommend the authors update Figure 1 itself if needed.

Reply: Thank you for your suggestion. We have edited Figure 2 (previously Figure 1) accordingly.

Round 2

Reviewer 1 Report

Authors have made sufficient revision. No further comment.

Reviewer 2 Report

The manuscript has been substantially improved in accordance with the previous reviewer's comments. There are no further comments from me, for its publication.